# CONTRAQA: QUESTION ANSWERING UNDER CONTRADICTING CONTEXTS

## ABSTRACT

With a rise in false, inaccurate, and misleading information in propaganda, news, and social media, real-world Question Answering (QA) systems face the challenges of synthesizing and reasoning over *contradicting information* to derive correct answers. This urgency gives rise to the need to make QA systems robust to misinformation, a topic previously unexplored. We study the risk of misinformation to QA models by investigating the behavior of the QA model under contradicting contexts that are mixed with both real and fake information. We create the first large-scale dataset for this problem, namely CONTRAQA, which contains over 10K human-written and model-generated contradicting pairs of contexts. Experiments show that QA models are vulnerable under contradicting contexts brought by misinformation. To defend against such threat, we build a misinformation-aware QA system as a counter-measure that integrates question answering and misinformation detection in a joint fashion.

## 1 INTRODUCTION

A typical Question Answering (QA) system (Chen et al., 2017; Yang et al., 2019; Karpukhin et al., 2020; Lewis et al., 2020b) starts by retrieving a set of relevant *context documents* from the Web, which are then examined by a machine reader to identify the correct answer. Existing work equate Wikipedia as the web corpus. Therefore, all retrieved context documents are assumed to be clean and trustable. However, real-world QA faces a much noisier environment, where the web corpus is tainted with *misinformation*. This includes unintentional factual mistakes made by human writers and deliberate disinformation intended to deceive. Aside from human-created misinformation, we are also facing the inevitability of AI-generated misinformation. With the continuing progress in text generation (Radford et al., 2019; Brown et al., 2020; Lewis et al., 2020a), realistic-looking fake web documents can be generated at scale by malicious actors (Zellers et al., 2019).

The presence of misinformation — no matter deliberately created or not, no matter human-written or machine-generated — affects the reliability of the QA system by bringing in *contradicting* information in the context documents. Figure 1 shows a question and five context documents, which give contradicting answers to the question. Only one context document (in green) is factually correct, while the rest are human-written or machine-generated fake information. Faced with such contradicting contexts, even human readers must be familiar with "Super Bowl 50" or rely on Web search to invalidate these fake contexts. Although current QA models often achieve super-human performance under the idealized case of clean contexts, we argue that they may easily fail under the more realistic case of contradicting contexts, especially when they do not have the ability to identify fake information and reason over contradicting contexts.

We seek to study risks of misinformation to QA models by investigating how QA models behave under contradicting contexts that are mixed with both real and fake information. Since there is no public QA dataset that marks and introduces contradicting contexts, we construct one ourselves: CONTRAQA, a large-scale dataset that specifically serves this need. Our dataset is constructed on top of SQuAD 1.1 (Rajpurkar et al., 2016). For each context paragraph $P$ in SQuAD, we create a fake version $P'$ by modifying information in $P$, such that: 1) certain information in $P'$ contradicts with the information in $P$, and 2) $P'$ is fluent, consistent, and looks realistic. We include both human-written and model-generated fake contexts in our dataset to simulate a realistic environment. For the human-written part, we ask Mechanical Turkers to write fake contexts by modifying

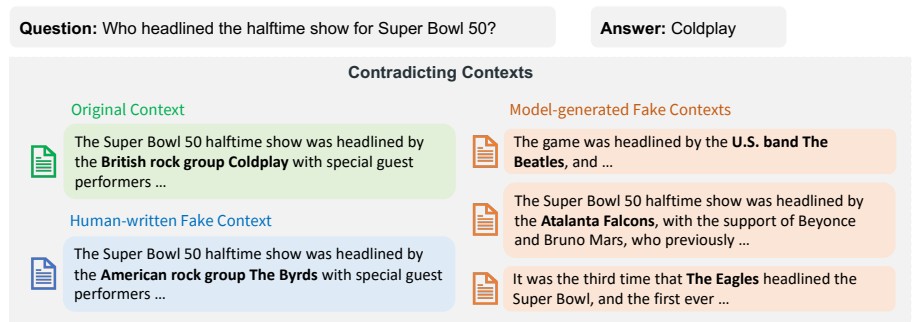

Figure 1: A data example from CONTRAQA, where contradicting information is in bold.

the original context. For the model-generation part, we propose a strong rewriting model, namely BART-FG, which can controllably mask and re-generate constituency spans in the original context to produce fake contexts. The original contexts, human-written and model-generated fake contexts are mixed together blindly and presented to QA model to answer a given question. A sample of CONTRAQA is shown in Figure 1, where the model is presented with multiple contradicting contexts to predict the answer for a given question. This stimulates a real-world situation where both real and fake information are retrieved as the context documents for open-domain QA. A robust QA model should be able to deal with misinformation and properly handle contradictory information.

Unfortunately, from extensive experiments, we find that existing QA models are vulnerable under contradicting contexts brought about by misinformation, regardless of whether the fake contexts are manually-written or model-generated. State-of-the-art QA models (Devlin et al., 2019; Liu et al., 2019; Joshi et al., 2020) all suffer from a significant drop in their exact match (EM) score by 20–30 when presented with contradicting contexts in CONTRAQA. Our analyses show that 1) our proposed context-rewriting BART-FG can create more deceiving misinformation than GPT-2 (Radford et al., 2019), and that 2) though machine-generated fake contexts are deceiving, they are generally not on par with human-written ones. We find that humans are still much better at making subtle and efficient edits to create contradicting information.

To defend against the potential threat of misinformation, we build a more robust QA system that integrates question answering with a misinformation discriminator in a joint fashion. Decisions made by discriminator assist QA models in identifying likely misinformation, which in turn improves the EM score by 17.2%, assuming access to a sufficient level of training data.

We plan to release CONTRAQA publicly, helping pave the way for building more robust QA systems. Although there are multiple types of misinformation in the real world, such as hoaxes, rumors, or false propaganda, we focus on the misinformation that brings conflicting information to the QA contexts. However, our work gives a general threat model for QA under misinformation, including an attack model that mixes both real and fake information in the QA context, and a defense model that combines question answering and misinformation detection. Followup research can easily build upon this threat model by studying how other types of misinformation mislead the QA systems. We summarize our contributions as follows:

• To the best of our knowledge, this is the first work to investigate QA under contradicting contexts.

• We construct a large-scale dataset CONTRAQA that includes contradicting contexts produced by both humans and neural models.

• We propose BART-FG, a novel framework that generates fake contexts by iteratively modifying constituency spans of the original context.

• To defend against the threat of misinformation, we propose a model that unifies question answering and misinformation discrimination.

## 2 RELATED WORK

**Adversarial Attacks in QA.** Adversarial attacks aim to trick QA models intentionally by perturbing their input contexts. These perturbations are expected to preserve the meaning of contexts while

degrading model performance. Ribeiro et al. (2018) make grammar perturbations such as replacing *What has* with *What's*. Other perturbations include adding distractor sentences (Jia & Liang, 2017; Wang & Bansal, 2018), modifying phrases (Maharana & Bansal, 2020), and human-in-the-loop edits (Bartolo et al., 2020). Although we also make modifications to the QA contexts, our setting differs from the task of adversarial attacks in QA. First, we aim to make the modified contents *contradict* with the original paragraph, while adversarial perturbations are expected to preserve the original meaning. Second, the goal of generating contradicting contexts is to study the impact of misinformation on QA models. In contrast, adversarial contexts aim to reveal the weaknesses of systems upon local changes that are imperceptible by humans.

**Improving Robustness for QA.** Our work aims to analyze vulnerabilities to develop more robust QA models. Current QA models demonstrate brittleness in different aspects. QA models often rely on spurious patterns between the question and context rather than learning the desired behavior. They might ignore the question entirely (Kaushik & Lipton, 2018), focus primarily on the answer type (Mudrakarta et al., 2018), or ignore the "intended" mode of reasoning for the task (Jiang & Bansal, 2019; Niven & Kao, 2019). QA models also generalize badly to out-of-domain (OOD) data (Kamath et al., 2020). For example, they often make inconsistent predictions for different semantically equivalent questions (Gan & Ng, 2019; Ribeiro et al., 2019). Notably, a concurrent work (Longpre et al., 2021) shows QA models are less robust to OOD data where the contextual information contradicts with the learned information. Different from that work, we propose a brand-new angle for QA robustness which studies the vulnerability of QA models under misinformation. Although our work and Longpre et al. (2021) both create contradictory contexts, we use them to study different problems. Moreover, we include both human-created and model-generated contradictory contexts, which are more flexible and diverse than their entity-based contradictions.

**Combating Neural-generated Misinformation.** While we are excited about the recent progress in neural text generation, they can be misused to generate realistic-looking and catchy hallucinations, such as fake news (Zellers et al., 2019) and fake online reviews (Garbacea et al., 2019). When produced at scale, neural-generated misinformation can pose threats to many NLP applications. We believe we are the first to study the risk of neural-generated misinformation to QA models and propose a misinformation-aware QA system as a countermeasure.

## 3  DATASET: CONTRAQA

We construct the CONTRAQA dataset as follows. We first select 10,026 unique context paragraphs from SQuAD 1.1, including all the 2,036 unique paragraphs from the validation set and 8,000 paragraphs randomly sampled from the training set. For each selected paragraph $\mathcal{C}^R$, we create a set of $N$ fake contexts $(\mathcal{C}_1^F, \cdots, \mathcal{C}_N^F)$ by modifying some information in $\mathcal{C}^R$, with the requirement that each fake context look realistic while containing contradicting information with $\mathcal{C}^R$.

We use two different ways to create fake contexts: 1) **via human edits**: we ask online workers from Amazon Mechanical Turk (AMT) to produce fake contexts by modifying the original context, and 2) **via BART-FG**: our novel generative model BART-FG, which iteratively masks and re-generates constituency spans from the original context to produce fake contexts.

### 3.1  MANUAL CREATION OF FAKE CONTEXTS

To solicit human-written deceptive fake contexts, we release 10K HITs (human intelligence tasks) on the AMT platform, where each HIT presents the crowd-worker with one context paragraph $\mathcal{C}^R$ we selected. We ask workers to modify the contents of the given paragraph to create a fake version, following the below guidelines:

• The worker should make *at least $M$ edits* at different places, where $M$ equals to one plus the number of sentences in the contexts $\mathcal{C}^R$.

• The worker should make at least one *long edit* that rewrites at least half of a sentence.

• The edits should modify key information to make it *contradict with the original*, such as time, location, purpose, outcome, reason, etc.

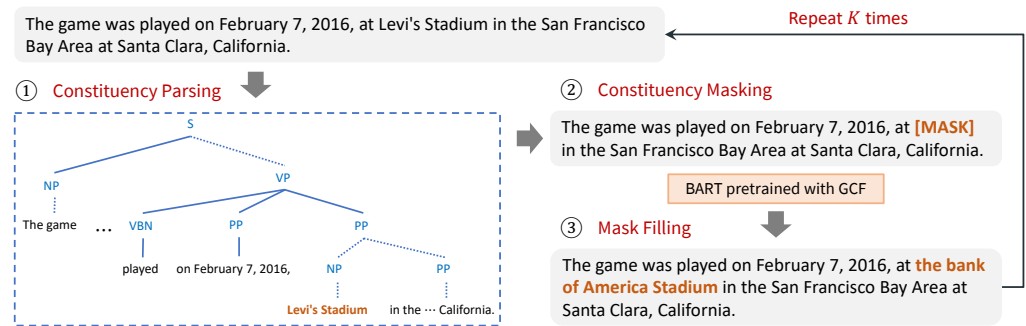

Figure 2: Overview of the BART-FG model, illustrated by an example sentence.

• The modified paragraph should be *fluent and look realistic*, without commonsense errors.

To select qualified workers, we restrict our task to workers who are located in five native English-speaking countries[1], and who maintain an approval rating of at least 90%. To ensure the annotations fulfil our guidelines, we give ample examples in our annotation interface with detailed explanations to help workers understand the requirements. The detailed annotation guideline is in Appendix B. We also hired three computer science major graduate students as human experts to validate a HIT's annotation. In the end, 104 workers participated in the task. The average completion time for one HIT is 5 minutes, and payment is $1.0 U.S. dollars/HIT. The average acceptance rate was 93.75%.

## 3.2 MODEL GENERATION OF FAKE CONTEXTS

Aside from human-written fake contexts, we also want to explore the threat of machine-generated fake contexts to QA. This source may be more of a concern than human-created misinformation, since they can easily be produced at scale. Recently introduced large-scale generative models, such as GPT2 (Radford et al., 2019), BART (Lewis et al., 2020a), and Google T5 (Raffel et al., 2020), can produce realistic-looking texts, but they do not lend themselves to producing controllable generation that only replaces only the key information with contradicting contents. Therefore, to evaluate the efficacy of realistic-looking neural fake contexts, we propose *BART Fake Contexts Generator (BART-FG)*, which produces both realistic and controlled generated text by iteratively modifying the original paragraph. As shown in Figure 2, for each sentence $S$ of the original paragraph, BART-FG produces its fake version $S'$ via a three-step process:

**1) Constituency Parsing**. We first apply constituency parsing to extract constituency spans from the input sentence to represent its syntactic structure. We use the off-the-shelf constituency parser from Joshi et al. (2018) in AllenNLP[2], which achieved 94.11 F1 on the Penn Treebank.

**2) Constituency Masking**. We then randomly select a constituency phrase[3] (a non-terminal in the parse tree) and replace it with a special mask token [MASK]. We choose to mask constituency phrases instead of random spans as: 1) constituents represent complete semantic units such as "Super Bowl 50", which avoids meaningless random phrases such as "Bowl 50"; and 2) constituents often represent important information in the sentence — such as time, location, cause, etc.

**3) Mask Filling**. We fill in the mask by generating a phrase different with the masked phrase. The mask is filled by the BART model fine-tuned on the Wikipedia dump with a new self-supervised task called *gap constituency filling*, introduced later.

The above pipeline is iteratively run for $K$ times to generate sentence $S'$ from $S$. We choose to make the edits iteratively rather than in parallel to model interaction between multiple edits. For example, in Figure 2, where the previous edit changed "Santa Clara" to "Atlanta", the next edit may change "California" into "Georgia" to make the contents more consistent and realistic.

---

[1]Australia, Canada, Ireland, United Kingdom, USA

[2]https://demo.allennlp.org/constituency-parsing

[3]Our considered constituency types: ADJP, ADVP, NP, PP, SBAR, SBARQ, SINV, VP, SQ, WHNP, WHPP

| # | Original Contexts | Contradicting Contexts |
|---|---|---|
| (1) | The game was played on February 7, 2016 at Levi's Stadium in the San Francisco Bay Area at Santa Clara, California. | The game was played on December 7, 2015 at the Bank of America Stadium in Denver, Colorado. |
| (2) | ... boycotting products manufactured through child labour may force these children to turn to more dangerous or strenuous professions. | ... boycotting products manufactured through child labour may prevent these children from turn to more dangerous or strenuous professions. |
| (3) | Tesla worked every day from 9:00 am until 6:00 pm or later. | Tesla worked every day but Sunday from 9:00 am until 6:00 pm or later. |
| (4) | The study suggests that boycotts are "blunt instruments with long-term consequences, that can actually harm the children involved." | The study did not find any major negative repercussions from boycotts, however, and found that boycotting is the best solution. |
| (5) | A key distinction between analysis of algorithms and complexity theory is that the former is devoted to ... , whereas the latter asks a more general question of ... | A key distinction between analysis of algorithms and complexity theory is that the latter is devoted to ... , whereas the former asks a more general question of ... |
| (6) | On the whole, Eisenhower's support of the nation's fledgling space program was officially modest until the 1957 Soviet launch of Sputnik, gaining the Cold War enemy enormous prestige around the world. | On the whole, Eisenhower's support of the nation's fledgling MK Ultra was officially terminated until the Cuban missile crisis, gaining the Cold War enemy enormous admiration in less developed nations. |

Table 1: Examples of original contexts and their corresponding contradicting versions from CON-TRAQA, where the edits are highlighted. Example (1) is from BART-FG. Examples (2)-(6) are from human. These examples represent six common ways of creating contradicting information.

**Gap Constituency Filling (GCF) Pre-Training.** To train the BART model to learn how to fill in a masked constituency phrase, we propose a novel pre-training task named *Gap Constituency Filling (GCF)*. For each article in the Wikipedia dump that consists of $T$ sentences $[S_1, S_2, \cdots, S_T]$, where each sentence is a word sequence $S_t = [w_1^t, \cdots, w_{|S_t|}^t]$, we construct the following training data for $t = 2, \cdots, T-1$:

Input: $S_1, S_{t-1}, w_{1:a-1}^t, [\mathtt{MASK}], w_{b+1:|S_t|}^t, S_{t+1}$          Output: $w_{a:b}^t = [w_a^t, \cdots, w_b^t]$

where the output represents a masked constituency span that starts with the $a$-th word and ends with $b$-th word. The input is the concatenation of the first sentence $S_1$, the previous sentence $S_{t-1}$, the current sentence $S_t$ with one constituency being masked, and the subsequent sentence $S_{t+1}$. The BART model is fine-tuned to predict the output given the input on the entire Wikipedia dump. This task trains the BART model to predict the masked constituent, given both global contexts ($S_1$) and local contexts ($S_{t-1}, S_{t+1}$).

## 3.3 DATA STATISTICS AND ANALYSIS

We collect 10,026 unique context paragraphs from the SQuAD dataset (8,000 from the train set, 2,026 from the dev set). For each paragraph $\mathcal{C}^R$, we create four fake contexts $\{\mathcal{C}_1^F, \cdots, \mathcal{C}_4^F\}$, where one fake context is written manually and the other three are generated via BART-FG. Afterward, each paragraph is paired with its corresponding question–answer pairs. Therefore, each data sample in CONTRAQA is a tuple of $(\mathcal{Q}, \mathcal{A}, \mathcal{C} = \{\mathcal{C}^R, \mathcal{C}_1^F, \cdots, \mathcal{C}_4^F\})$, where $\mathcal{C}$ is the contradicting contexts that are mixed with one original context and four fake contexts. Since each context $\mathcal{C}$ is paired with multiple $(\mathcal{Q}, \mathcal{A})$ in SQuAD, we finally obtain 36,447 and 10,218 data samples for the train and dev set, respectively. Since the SQuAD test set is not released, we only create contradictory examples for the full SQuAD dev set. For a fair comparison, we compare the performance of QA models between the original and contradictory versions of the SQuAD dev set.

Table 1 shows six original contexts with their corresponding contradicting contexts, which represent six common types of modifications, explained in the following:

(1) **Entity Replacement**: replacing entities (*e.g.*, person, location, time, number) with other entities with the same type, a common type of modification for both human edits and BART-FG.

(2) **Verb Replacement**: replacing verb or verb phrase with its antonymic meaning, *e.g.*, "force these children to" → "prevent these children from".

(3) **Adding Restrictions**: create contradiction by inserting additional restrictions to the original content, *e.g.*, "every day" → "every day but Sunday".

(4) **Sentence Rephrasing**: rewrite the whole sentence to express a contradicting meaning, exemplified by (4). This is common in human edits but rarely seen in model-generated contexts, since this requires deep reading comprehension.

(5) **Disrupting Orders**: make a contradiction by disrupting some property of the entities; *e.g.*, example (5) switches the property of "analysis of algorithms" and "complexity theory".

(6) **Consecutive Replacements**: humans are better in making consecutive edits to create a contradicting yet coherent sentences, exemplified by (6).

## 4 MODELS AND EXPERIMENTS

We now study how extractive QA models behave under such contradicting contexts. Extractive QA aims to answer a given question $\mathcal{Q}$ by selecting a span from a set of context paragraphs $\mathcal{C}$. We apply this to contradicting contexts by proposing two settings as follows:

**1. Contra-QA.** In this setting, QA is conducted under contradicting contexts, *i.e.*, $\mathcal{C} = \{\mathcal{C}^R, \mathcal{C}_1^F, \cdots, \mathcal{C}_N^F\}$, where $N = 4$ in our CONTRAQA dataset. We shuffle the context paragraphs so that the model does not know which context is real. Following Chen et al. (2017), a QA module (a.k.a. passage reader) is applied to each of the paragraph $C_i$ to select the best answer span $A_i$ and provides its confidence score $S_i$. Afterward, the system returns the answer with the highest (normalized) span score.

**2. Contra-QA (w/ Detection).** In order to mitigate the harm of misinformation, we propose a misinformation-aware QA framework that integrates question answering with fake context detector. For each context paragraph $\mathcal{C}_i$, a fake context detector outputs its *trust score* $R_i$; *i.e.*, the confidence score that information in $\mathcal{C}_i$ is trustable. We then combine the trust score $R_i$ with the confidence score $S_i$ for the best answer span in $\mathcal{C}_i$ via linear interpolation: $P_i = \lambda \cdot S_i + (1 - \lambda) \cdot R_i$, where $\mu \in [0, 1]$ is a hyperparameter which we set as 0.5. Finally, we select the answer span with the best $P_i$ across all context paragraphs as the final prediction.

The above settings are compared with two settings without contradictory contexts. **1) SQuAD**: the traditional QA setting in which only the real context is given. **2) SQuAD + Random Ctx.**: the real context is paired with $N$ randomly sampled other contexts irrelevant to the question. This helps to differentiate whether the model is really distracted by the contradicting contexts or simply by the fact that there are other contexts.

We consider four state-of-the-art QA models with public code that achieved strong results on the public leader board of SQuAD: *BERT-base* (Devlin et al., 2019), *RoBERTa-base*, *RoBERTa-large* (Liu et al., 2019), and *Span-BERT* (Joshi et al., 2020). We use their implementations from the Hugging Face library, fine-tuned on the SQuAD-1.1 training set. We use the standard Exact Match (EM) and $F_1$ metrics to measure QA performance. We train the fake context detector by fine-tuning the RoBERTa-large model to differentiate real and fake paragraphs (binary classification) using a moderate level of labeled real/fake paragraphs in the CONTRAQA training set. The classifier achieves an detection accuracy of 80.57% with 10,000 training examples.

### 4.1 MAIN RESULTS

In Table 4, we show the performance of different QA models on the CONTRAQA test set. We have two major observations.

**Misinformation can easily mislead QA models.** When moving from clean SQuAD contexts to contradicting contexts, all four QA models suffer from large performance drops between 23.4 (BERT-base) and 25.91 (RoBERTa-large) in absolute EM value, and drops between 40.67% and 42.24% in relative EM value. This reveals the serious impact of misinformation and its potential

| Model | SQuAD | SQuAD + Random Ctx. | Contra-QA | Contra-QA + Detector |
|---|---|---|---|---|
| | EM / $F_1$ | EM / $F_1$ | EM / $F_1$ | EM / $F_1$ |
| BERT-base (Devlin et al., 2019) | 80.94 / 88.07 | 76.59 / 82.82 | 57.54 / 67.84 | 66.76 / 75.34 |
| RoBERTa-base (Liu et al., 2019) | 85.01 / 91.46 | 76.78 / 81.75 | 60.95 / 70.24 | 70.29 / 80.01 |
| RoBERTa-large (Liu et al., 2019) | 87.25 / 93.53 | 80.01 / 84.85 | 61.34 / 71.33 | 71.92 / 81.30 |
| Span-BERT (Joshi et al., 2020) | 84.49 / 91.69 | 75.30 / 80.08 | 59.31 / 69.68 | 69.73 / 79.05 |

Table 2: QA performance for four different models under the four different settings.

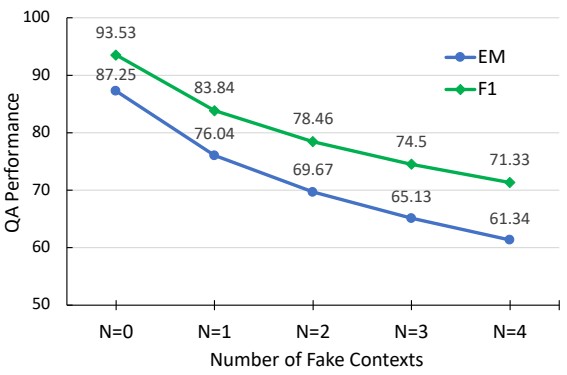

Figure 3: The QA performance for RoBERTa-large model with different number of fake contexts $N$.

Figure 4: Error analysis, showing the QA model tends to be deceived by the fake context generated by which method?

threat to current QA systems. *As QA models are not trained to differentiate fake contexts, they can be easily mislead by misinformation.*

**QA models are mainly distracted by contradictions brought by misinformation.** By pairing the original context with four randomly sampled other contexts (SQuAD + Random Ctx.), we only observe an average EM drop of 9.4%, showing that the QA models are relatively robust when using random distractor contexts. However, when using the contradictory contexts as the distractor (Contra-QA), we observe an average 41.21% drop in EM. The results show that *the models are largely distracted by the contradictory contexts rather than by the presence of additional contexts.*

**Misinformation-aware QA models are more robust under contradicting contexts.** When the RoBERTa-large QA model is equipped with the fake context detector trained with 10,000 examples, the EM score improves from 61.34% to 71.92% (+17.2%). Similar improvements are observed for other QA models. This result shows that the decisions of the fake context detector can assist QA models in identifying likely misinformation, thus improving the QA performance under contradicting contexts. However, the performance is still much lower than the results under clean context (87.25% EM), due to the imperfect accuracy of the detector. Moreover, training an effective detector is often difficult in the real world, as it requires the time-consuming and costly labeling of large amounts of in-domain real and fake contexts as training data. Therefore, *how to effectively fight against the threat of misinformation in QA remains challenging and deserves future attention.*

## 4.2 IMPACT OF THE NUMBER OF FAKE CONTEXTS

Given the contradicting contexts $\mathcal{C} = \{\mathcal{C}^R, \mathcal{C}_1^F, \cdots, \mathcal{C}_N^F\}$ (cf Figure 3), we plot the EM and F1 for the RoBERTa-large model with $N = 0, 1, 2, 3, 4$. The results show a linear downward trend of both EM and F1 as $N$ increases. Therefore, *misinformation may have more severe impact on QA systems when they are produced at scale.* With the availability of pretrained text generation models, producing fluent and realistic-looking contexts now has little marginal cost. This brings an urgent need to effectively defend against neural-generated misinformation.

| Setting | Generation Method | Edit (%) | Performance (EM / F1) | | | |
|---|---|---|---|---|---|---|
| | | | N=1 | N=2 | N=3 | N=4 |
| Contra-QA | Human | 19.19 | 72.02 / 81.88 | — | — | — |
| | BART-FG (K=1) | 24.82 | 81.40 / 88.22 | 78.37 / 85.58 | 75.84 / 83.36 | 74.27 / 81.82 |
| | BART-FG (K=2) | 37.93 | 78.85 / 86.01 | 74.13 / 81.79 | 71.07 / 79.11 | 68.86 / 77.04 |
| | BART-FG (K=3) | 46.07 | 77.51 / 84.76 | 72.20 / 80.05 | 68.68 / 76.94 | 66.64 / 75.11 |
| | GPT2-FG | 54.05 | 83.03 / 88.93 | 81.46 / 87.27 | 80.46 / 86.19 | 79.73 / 85.33 |
| SQuAD | | — | 87.25 / 93.53 | | | |
| SQuAD + Random Ctx. | | — | 84.97 / 90.58 | 83.08 / 88.38 | 81.32 / 86.42 | 80.01 / 84.85 |

Table 3: Evaluations of QA performance for different methods of generating fake contexts.

### 4.3 WHICH IS MORE DECEIVING: HUMAN- OR MODEL-GENERATED MISINFORMATION?

We further investigate which is more deceiving to QA models: human or neural misinformation? To study this, based on the real contexts $\mathcal{C}^{\mathcal{R}}$ in the CONTRAQA test set, we generate their fake contexts $\{\mathcal{C}_1^F, \cdots, \mathcal{C}_N^F\}$ using different methods and then evaluate the QA performance of the RoBERTa-large model. The methods we consider are human and BART-FG; where in the case of BART-FG we consider three variants in which the number of iterations $K$ is set as 1, 2, and 3, respectively.

Table 3 shows the QA performance for different methods. We introduce a metric called *average edit distance percentage* (Edit(G)) to measure the average number of edits a generation method $G$ needs to make to the original contexts in order to generate the fake paragraph, defined as follows: $\text{Edit}(G) = \sum_{i=1}^{M} \left| \text{edit\_distance}(G(\mathcal{C}_i^R), \mathcal{C}_i^R)/\text{length}(\mathcal{C}_i^R) \right|$, where $\mathcal{C}_i^R$ is the $i$-th real context in the test set, and $G(\mathcal{C}_i^R)$ is the fake context generated by method $G$ for $\mathcal{C}_i^R$. This metric measures the relative edit distance between the true and fake context, taking average in the test set. From the results in Table 3, we make two major observations:

**Humans can create more misleading contradictions with fewer edits.** Table 3 shows that when pairing the real context with the human-written fake context, the models underperform (an EM 72.02%) compared against pairings with model-written fake contexts. Interestingly, the average edit distance percentage for humans is also the lowest (19.19%) among all methods. This indicates that humans create more challenging fake contexts that trick QA models with fewer edits. From our observations in Table 1, humans make more subtle and deceiving edits to create contradictions, such as switching "former" and "latter" (Example 4), and changing "every day" to "every day but Sunday" (Example 3). Such edits requires a deep level of reading comprehension not currently achieved by text generation models.

Error analysis also reveals that human-created contradictions are more deceiving. As shown in Figure 4, for each real context, we pair it with five fake contexts produced by five different methods. We then analyze the source (which fake context) of the incorrect answer when the RoBERTa-large QA model makes an error. If all five methods create equally deceiving fake contexts, we expect to observe a uniform distribution. However, the distribution in Figure 4 shows that the most (31.8%) wrong answers are extracted from the human-created fake context.

**BART-FG creates more deceiving contradictions at the cost of more edits.** As the number of iterations $K$ increases, the BART-FG model makes more edits to the original contexts, as reflected by the increasing Edit in Table 3. The resultant QA performance also falls, as more contradicting information is likely to appear when more edits are made. Therefore, generation models can produce more deceiving fake contexts by making more edits to the real contexts (certainly worrying from a dual-use perspective). However, we will show in Section 4.5 that more edits make the generated texts less realistic and more easy to detect.

### 4.4 BART-FG VERSUS GPT2

Is GPT2 also good at generating deceiving fake contexts? To investigate this, we introduce a baseline — namely GPT2-FG— to compare against our proposed BART-FG model. GPT2-FG applies the pretrained GPT2-large model from Hugging Face to generate the rest of the contexts given the first 20% of the real contexts $\mathcal{C}^{\mathcal{R}}$ as the prompt.

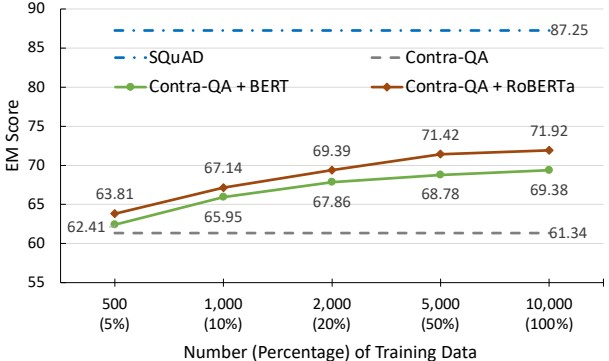

Figure 5: QA performance when applying the fake context detector trained with different amount of data.

| Setting | Detector Accuracy | EM score before → after adding detector |
|---------|-------------------|------------------------------------------|
| Human | 72.86% | 72.02 → 78.92 |
| (K=1) | 78.04% | 81.40 → 85.26 |
| (K=2) | 84.17% | 78.85 → 85.95 |
| (K=3) | 90.10% | 77.51 → 85.91 |

Figure 6: Independent evaluation of the detector accuracy for fake contexts generated by different methods and the benefits to QA models. The second to fourth rows denote the BART-FG model with different parameter $K$.

We find that BART-FG is able to create more contradictions with less edits compared with the GPT2-FG baseline (Table 3). We attribute this to the iterative modification strategy of BART-FG, which selectively replaces text spans that convey key information. In contrast, while GPT2-FG generates the whole passage without any explicit control except for the given prompt. This often makes the generated contents deviate the original topic, yielding fewer contradictions. The error analysis in Figure 4 also shows that GPT2-FG generates the least deceiving fake contexts compared with human and BART-FG.

## 4.5 Evaluation of Misinformation Detection

Finally, we evaluate effectiveness of integrating a misinformation detector of varying model architecture, and their sensitivity to training data size. Figure 5 shows the EM score achieved by the RoBERTa-large QA model after incorporating different detectors. While preliminary, the results show that we can train an effective detector only when we have sufficient number of in-domain labeled real/fake contexts. The benefit of the detector becomes quite limited given with insufficient training data (*e.g.*, +4.0% with 500 training samples). This reveals the difficulty of defending against misinformation in the real world: ideally a good detector helps, but we usually do not have large-scale in-domain labeled data to train an effective detector.

Through a separate evaluation for different types of fake contexts, we further show in Figure 6 that human-written misinformation has the lowest detection accuracy, showing that they are more difficult to detect than the machine-generated misinformation. This further validates the observation in Section 4.3 that humans can create subtle misinformation that require a high-level understanding. For BART-FG, we find a trade-off between *contradiction power* and *realism*. As $K$ increases, the model makes more edits, creating more contradicting information, which lowers the EM score for QA. However, more edits will make the generated fake contexts less realistic, leaving more of a "trace" for the detector to track, thus increasing the fake detection accuracy. When a detector is applied, these two factors cancel out and give us a similar EM score of around 85.

## 5 Conclusion and Future Work

We study the potential threat of misinformation on question answering models by creating a large-scale dataset CONTRAQA, containing over 10K human-written and model-generated contradicting contexts that are mixed with both real and fake information. Our studies reveal that QA models are indeed vulnerable under contradicting contexts. While integrating a misinformation detector into the QA system mitigates the problem, this solution requires the labeling of large-scale real/fake paragraphs which may not be feasible nor generalizable. We believe urgent further work is required to study this problem under the more realistic open-domain QA setting, to propose more effective counter-measures to build a robust misinformation-aware QA system. We make our dataset and codes publicly available to further this important agenda.

## ETHICS STATEMENT

We plan to publicly release the CONTRAQA dataset and open-source the code and model weights for our BART-FG model. We note that open-sourcing the BART-FG model may bring the potential for deliberate misuse to generate disinformation for harmful applications. Since our CONTRAQA dataset contains large-scale human-written and model-generated fake contexts, it can also be misused to generate disinformation. We deliberated carefully on the reasoning for open-sourcing and share here our three reasons for publicly releasing our work.

First, the danger of BART-FG in generating disinformation is limited. Disinformation is a subset of misinformation that is spread deliberately to deceive. Although we utilize the innate "hallucination" ability of current pretrained language models to create misinformation, our model are not specialized to generate harmful disinformation such as hoaxes, rumors, or false propaganda. Instead, our model focuses on generating conflicting information by iteratively editing the original passage to test the ability of QA models to handle contradictory information.

Second, our model is based on the open-sourced BART model, which makes our model easy to replicate even without the released code. Given the fact that our model is a revised version of an existing publicly available model, it is unnecessary to conceal code or model weights.

Third, our decision to release follows the similar stance of the full release of another strong detector and state-of-the-art generator of neural fake news: Grover (Zellers et al., 2019)[4]. The authors claim that to defend against potential threats, we need threat modeling, in which a crucial component is a strong generator or simulator of the threat. In our work, we build an effective threat model for QA under misinformation. Followup research can build on our model transparency, further enhancing the threat model.

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

## A    FIGURE ILLUSTRATION OF EXPERIMENTAL SETTINGS

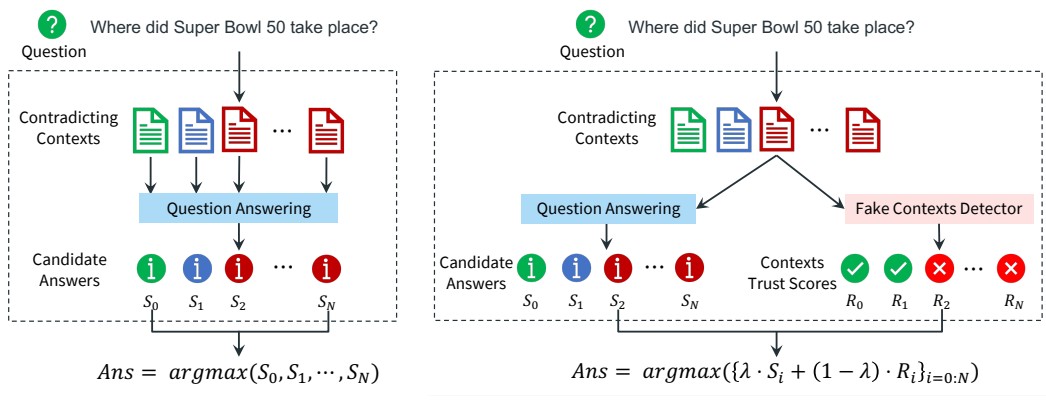

(a) QA under Contradicting Contexts    (b) QA under Contradicting Contexts + Fake Context Detector

Figure 7: The *Contra-QA* setting (a) and the *Contra-QA w/ Detector* setting (b).

## B    HUMAN ANNOTATION GUIDELINE

### B.1    JOB DESCRIPTION

Given a paragraph from Wikipedia, modify some information in the paragraph to create a fake version of it. Here are the general requirements:

• You should make *at least $M$ edits* at different places, where $M$ is determined by the length of the passage and will show on the screen when you annotate each passage.

• You should make at least one *long edit* that rewrites at least half of a sentence.

• The edits should modify key information to make it *contradict with the original*, such as time, location, purpose, outcome, reason, etc.

• The modified paragraph should be *fluent and look realistic*, without commonsense errors.

For example, given the following passage:

> Super Bowl 50 was an American football game to determine the champion of the National Football League (NFL) for the 2015 season. The American Football Conference (AFC) champion Denver Broncos defeated the National Football Conference (NFC) champion Carolina Panthers 24-10 to earn their third Super Bowl title. The game was played on February 7, 2016, at Levi's Stadium in the San Francisco Bay Area at Santa Clara, California. As this was the 50th Super Bowl, the league emphasized the "golden anniversary" with various gold-themed initiatives, as well as temporarily suspending the tradition of naming each Super Bowl game with Roman numerals (under which the game would have been known as "Super Bowl L"), so that the logo could prominently feature the Arabic numerals 50.

Modify some key information of it to create the following fake version:

> Super Bowl 50 was the 48th Super Bowl Game to determine the champion of the National Football League (NFL) for the 2015 season. The American Football Conference (AFC) champion San Francisco 49ers defeated the National Football Conference (NFC) champion Carolina Panthers 24-10 to earn their third Super Bowl title. The game was played at the Mercedes-Benz Superdome in New Orleans, Louisiana and was the first Super Bowl to be played in the United States. As this was the NFL's 48th Super Bowl, the league emphasized the "golden anniversary" with various gold-themed initiatives, as well as temporarily suspending the tradition of naming Super Bowls with Roman numerals (under which the game would have been known as "Super Bowl L"), so that the game would be known as the "Super Bowl of the Century".

## B.2 DETAILED REQUIREMENTS

Here we give an example of modification as follows.

Detailed annotation instructions are as follows.

**1) At least make N edits at different places.** In the above example, there are a total of 5 edits:

- "an American football game" → "the 48th Super Bowl Game"
- "Denver Broncos" → "San Francisco 49ers"
- "on February 7, 2016, at Levi's Stadium in the San Francisco Bay Area at Santa Clara, California." → "Mercedes-Benz Superdome in New Orleans, Louisiana and was the first Super Bowl to be played in the United States."
- "the 50th" → "the NFL's 48th"
- "so that the logo could prominently feature the Arabic numerals 50." → "so that the game would be known as the "Super Bowl of the Century."

**2) There should be at least one long edit.** Among all your edits, there should be at least one long edit, which rewrites the whole sentence or at least half of the sentence.

In the above example, the long edit is: "on February 7, 2016, at Levi's Stadium in the San Francisco Bay Area at Santa Clara, California." → "Mercedes-Benz Superdome in New Orleans, Louisiana and was the first Super Bowl to be played in the United States."

**3) The edits should create contradicting information.** After your edits, the original passage and the modified passage should have contradicting information. One way to test it is that: when you ask questions about your modified information, the original passage and the modified passage gives contradicting answers.

For example: after you edit "Denver Broncos" to "San Francisco 49ers", the original and modified passages are shown in the Figure below:

Original Text:

```
The American Football Conference (AFC) champion Denver Broncos
defeated the National Football Conference (NFC) champion Carolina
Panthers 24-10 to earn their third Super Bowl title.
```

Modified Text:

```
The American Football Conference (AFC) champion San Francisco
49ers defeated the National Football Conference (NFC) champion
Carolina Panthers 24-10 to earn their third Super Bowl title.
```

When you ask the question: "Which NFL team won Super Bowl 50?", the original passage gives you the answer "Denver Broncos", and the modified passage gives you the answer "San Francisco 49ers". This is a contradiction.

Another example is the following edit: "so that the logo could prominently feature the Arabic numerals 50." → "so that the game would be known as the "Super Bowl of the Century".

Original Text:

```
...  the league emphasized the "golden anniversary" with various
gold-themed initiatives, as well as temporarily suspending the
tradition of naming each Super Bowl game with Roman numerals
(under which the game would have been known as "Super Bowl L"),
so that the logo could prominently feature the Arabic numerals
50.
```

Modified Text:

```
...  the league emphasized the "golden anniversary" with various
gold-themed initiatives, as well as temporarily suspending the
tradition of naming each Super Bowl game with Roman numerals
(under which the game would have been known as "Super Bowl L"), so
that the game would be known as the "Super Bowl of the Century".
```

When you ask the question: "Why the league suspended the tradition of naming Super Bowls with Roman numerals?" the original passage and the modified passage also give you contradicting answers.

However, the following passage does **NOT** create any contradiction, because the modified information is just a paraphrasing of the original information.

Original Text:

```
The American Football Conference (AFC) champion Denver Broncos
defeated the National Football Conference (NFC) champion Carolina
Panthers 24-10 to earn their third Super Bowl title.
```

Modified Text:

```
The American Football Conference (AFC) champion Denver Broncos
defeated the National Football Conference (NFC) champion Carolina
Panthers 24-10 to win the Super Bowl.
```

**4) The edits should modify important information in the passage.** Your edits should focus on important information in the passage, *i.e.*, points that people are usually interested in and would usually ask about. For example, time, location, purpose, outcome, reason, etc. Please avoid editing trivial and unimportant details.

For example, the following trivial edit is not supported:

Original Text:

```
the game would have been known as "Super Bowl L"...
```

Modified Text:

```
the game would have been known as "Super Bowl H"...
```

**5) The modified passage should look "realistic".**  The final modified passage should look "realistic". Don't make obvious logic or commonsense mistakes to make the reader easily know that this is a fake passage by simply going through it.

For example, the following edit is not supported.

Original Text:

```
The game was played on February 7, 2016, at Levi's Stadium in the
San Francisco Bay Area at Santa Clara, California.
```

Modified Text:

```
The game was played on February 7, 2016, at Levi's Stadium in the
San Francisco Bay Area at New York City, California.
```

People can easily tell the modified passage is fake since everybody knows that New York is not a city in California.

## B.3 ANNOTATION INTERFACE

The original passage is shown on the left for your reference, you should modify the passage in the text box on the right to make the fake passage. After you finished the edits, Click "Submit".

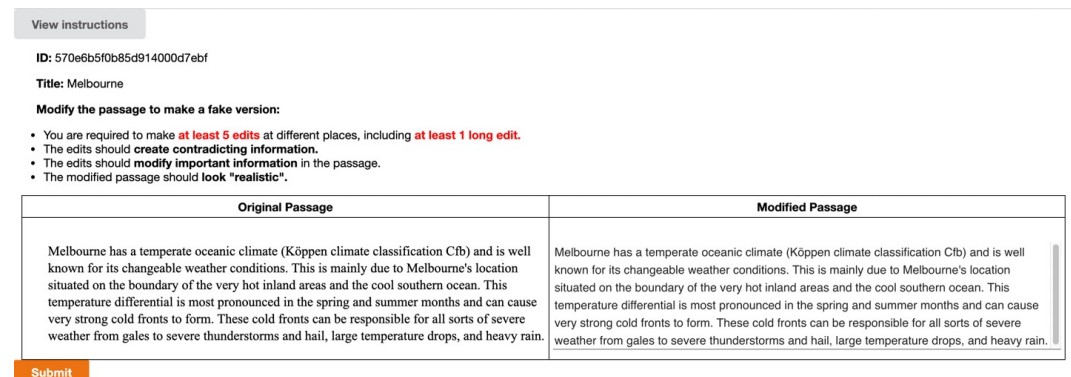

Figure 8: The annotation interface in the Amazon Mechanical Turk.

