# OpenReview forum: "ContraQA: Question Answering under Contradicting Contexts"
_ICLR.cc/2022/Conference — ICLR 2022 Submitted_

### Official Review · Reviewer_4eU9 · 2021-10-23

**Correctness:** 2
**Technical Novelty And Significance:** 1
**Empirical Novelty And Significance:** 1
**Recommendation:** 3
**Confidence:** 4

**Main Review:**

1. What is the goal of the paper? If the goal is to present a dataset that is about misinformation -- it is not simply enough to create a contradictory passages. This is just another way of distracting a QA model but without having any reason for it to believe the information is "fake". In order for something to be "fake", there has to be some ground-truth known. The experiment called Contra-QA appears to be flawed given what it was supposed to check. How is a model expected to learn which passage is real? It has to be grounded in something that it can rely on for evidence isn't it? Would human beings know if something is fake unless there are also aware of what a "trustworthy" source says? Perhaps the authors can elaborate further (in case I have badly misunderstood the work).

2. Similarly, if you add contradictory information to passages for a span-based QA model its no surprise it gets confused. Neither are those models trained to not respond in the presence of contradictory information nor are they being told which passage is real (the trust-score is truly not a trust-score -- it is just the output of a model that  frankly appears to be guessing because it has no way of knowing what is trustworthy!).

3. What could perhaps have been interesting is to also see if a model could "detect" contradictions and says, that it should not answer. This is a model that you can easily train with this data and perhaps the only thing meaningful I can think of doing with this dataset without having any access to methods that tell the system what is "real".

I found the methods for generating contradictory passages novel and interesting and could find more general use in other tasks related to dataset augmentation. That limited contribution, however is not enough to accept this paper in its current form.

**Summary Of The Paper:**

This paper releases a new dataset with human and machine generated contradictory contexts for QA pairs from SQuAD 1.1. Amazon Mechanical Turk Workers are shown a paragraph and are asked to make edits such that it contradicts the original paragraph with respect to elements such as time, outcome, purpose, location, etc. In addition, the authors fine tune BART on a collection of masked constituent parses of Wikipedia sentences and it is then trained to fill the mask with an alternative phrase.  To automatically generate contradictory contexts, the authors use this fine tuned the BART model on the masked constituency parse of paragraph sentences. A dataset of 10,000 paragraphs from SQuAD  are transformed (once by mechanical Turk workers, and the rest by three different transformations by the BART Model). The paper presents experiments on this dataset for the task of QA -- specifically, machine reading comprehension. In one experiment, the QA system is first trained to predict which of the 5 (1 real + 4 contradictory) is correct. Then an off-the-shelf span based passage reader returns spans as answers. In the second experiment, the performance of QA models is compared on the unmodified SQuAD dataset as well as a version where a distracting passage is also added to context (by randomly choosing a different passage). The authors experiment using BERT, ROBERTA and SPAN-BERT and report a drop in performance in both experimental settings. In addition experiments reveal that models return worse performance on the subset of the data created by human workers ( perhaps unsurprising). Additional studies on the nature of edits have also been presented.

Overall a well written and easy to read paper. However, I am not sure I am clear about the goals of the paper -- I elaborate further in the rest of the review.




**Summary Of The Review:**

The paper's experiments do not back up the claims of reasoning for misinformation. It is about reasoning with contradictory information and its no surprise QA models dont know what to answer when that happens. Why is this surprising? What is the goal of the work.
I apologize if I have badly misunderstood the work and I'd encourage the authors to discuss these comments in the rebuttal.

---

### Official Review · Reviewer_g2wC · 2021-10-31

**Correctness:** 3
**Technical Novelty And Significance:** 2
**Empirical Novelty And Significance:** 3
**Recommendation:** 5
**Confidence:** 4

**Details Of Ethics Concerns:**

Authors already discussed the potential risk of publishing the tools to generate the dataset. Reviewer agrees that they are not adding significant new risk given the proliferation of similar tools.

**Main Review:**

Strength.

- It calls our attention to a very realistic problem, that misinformation could affect the QA systems, which a lot of people may blindly rely on.
- Its pledge to share the data. It will help future studies in the same direction.
- The description of data preparation and characteristics is very detailed and convincing.
- The experiments are very detailed.
- The paper itself is well-written and easy to follow.

Weakness: Let me order them from most to least important.

- It's not clear how we can tell the truth from the contradictory ones based on content only, and how the RoBERTa-based classifier did it (to 80%). For example, a common way to create the contradictory context is to replace one or more of the time/location or named entities ("San Francisco" -> "Atlanta"), or changing modifiers ("all the time" -> "all the time except Sunday"). It would be impossible for humans to tell, without strong background knowledge. Reviewer is very curious about what the classifier had learned, which is not discussed in the paper.

- Leveraging the trustworthiness of the source, which seems like an apparent solution to Reviewer, is not discussed in the paper, at least not in the related work. For example, a quick search in Google Scholar led me to https://journals.sagepub.com/doi/abs/10.1177/0165551513478893.

- How Gap Constituency Filling (GCF) Pre-Training produce a *contradictory* filler. The pre-training will result in the most likely filler. Avoiding the original text will make sure it's different, but not necessarily contradictory. How did the authors make sure it's contradictory? It's worth explaining.

- In section 3.1 the requirement to human labelers, the authors insist that "The worker should make at least M edits at different places, where M equals to one plus the number of sentences in the contexts". Reviewer is not clear why we need this specific minimum".



**Summary Of The Paper:**

The authors studied how contradictory information affects the accuracy of QA systems.  They created a new dataset of ~10k from SQuAD with added contradictory paragraphs (context). The contradictory data was generated by two different ways. The authors employed Amazon Mechanical Turks raters to rewrite the original context. They also designed a BART-FG model to automatically produce such context by replacing spans with model generated value.

The authors also proposed a way to help QA systems avoid contradictory information. A RoBERTa-based model was used to classify a context into trustworthy or not, with an accuracy around 80%. The trustworthiness score is then used to weigh the final result, together with existing scores of quality/confidence.

Evaluation was done with the new dataset. It showed that 1) adding contradictory information hurts QA performance badly, 2) the RoBERTa-based classifier can help regain some of the loss, but not all of them. The authors also measured the effectiveness of contradictory information creation, where human takes the top place by producing the least altered context with largest effect on final outcome.

The authors promised to share the dataset and the source code / weights of the proposed models. They further discussed the potential ethical impact of releasing the data, arguing that it's net beneficial.


**Summary Of The Review:**

The paper proposed a novel study of QA system robustness under contradictory information. It's well written and provides a new dataset. However the discussion around how a classifier could tell truth from noise and why don't we leverage source authority are missing. Some smaller issues exist too. Reviewer would like to see the two bigger questions answered.

---

### Official Review · Reviewer_nPFz · 2021-11-02

**Correctness:** 4
**Technical Novelty And Significance:** 3
**Empirical Novelty And Significance:** 3
**Recommendation:** 6
**Confidence:** 3

**Main Review:**

**Strengths**:

- To my belief, this is the first work that sets up contradicting contexts for closed-domain QA. It is very important for NLP and QA community to have more challenging evaluation benchmarks to understand how well models "generalize".
- The work delineates an interesting study of how current QA systems perform when given contradicting contexts for reference, and also proposes a system that performs well at discriminating misinformation introduced by the ContraQA task.
- The insight that human generated contradictions were "stronger" (were relatively more capable to fool QA systems) than Neural generations highlights the scope for developing better adversarial rewriting models.
- The paper is fairly easy to follow

**Weaknesses**:

The work leaves a few things to be desired:
- The work only considers SQuAD which is demographically one of the most skewed datasets (Gor et. al. 2021). It would been interesting to see if this framework generalizes to NQ or newer and more challenging table-text based QA tasks like HybridQA (Chen et. al. 2020).
- Though the error analysis is interesting, I find it a bit shallow. The work doesn't throw light on the discriminating features of human written and neurally generated contradicting contexts. Knowing this can help in corporate certain type of edits in neural models, or help humans write better contradicting contexts in some way.

**Followup Questions:**

1. Another interesting baseline to compare with ContraQA would be SQuAD + N Most similar (tf-idf) context passages (instead of random). Had that been tried ?
2. The fourth guideline for fake context creation by humans: "The modified paragraph should be fluent and look realistic, without commonsense errors.", How was this objectively evaluated?
3. From interpretability point of view, what helps the discriminator network filter off the contradicting contexts? Is it speculated to be just the Wikipedia pre-training? If so, how do you expect the miss information at source to play a role in confounding the QA systems?

**Nits:**
1. § 4. Contra-QA (w/ Detection): Do you mean $\lambda$ instead of $\mu$ ?


**Summary Of The Paper:**

The work investigates closed-domain Question Answering under contradicting contexts by introducing a new task ContraQA—an extension of SQuAD1.1—which includes contradicting contexts for the SQuAD articles, produced by both humans and neural models. The work also proposes a neural framework, BART-FG, to automatically generate these contradicting contexts by iteratively modifying constituency spans on the original context. Finally, the work gives a brief analysis on how SOTA QA systems perform on the new task, ContraQA, and proposes a misinformation detecting system which when unified with a Machine Reader performs significantly better than SOTA systems over ContraQA.

**Summary Of The Review:**

Putting together all the strengths and weaknesses I believe the NLP and QA community will benefit from the insightful outcomes of this work. However at the same time, it does leave things to be desired. Nonetheless, I am inclining to accept this work.

---

### Official Review · Reviewer_2VLR · 2021-11-03

**Correctness:** 3
**Technical Novelty And Significance:** 2
**Empirical Novelty And Significance:** 2
**Recommendation:** 3
**Confidence:** 4

**Details Of Ethics Concerns:**

The fact that the model is based on BART, and can be easily reproduced, is not a valid justification to release the trained model BERT-FG. Same goes with the point regarding the limited ability to generate disinformation. While every generative model can be used in theory to create fake contexts, this work describes a way of generative fake contexts, without providing (based on the results), any robust approach to mitigate the problem.

**Main Review:**

The purpose idea of the dataset is valuable, as well as the problem is proposing to address. However, there are some problems that do not allow me for acceptance.

Problem statement.
1)	A considerable part of the paper is dedicated to the generation of the so-called contradicting examples. However, the way in which they are generated is in line with other works (cited in the paper as well) where the original text is perturbed by modifying entities. What makes the examples in ContraQA contradicting examples? A definition is indeed required.

2)	The problem of misinformation has been explored in the literature, and many works treat the task as a fact checking problem. Because the literature regarding fact-checking is not considered at all in the paper, what makes the problem addressed in this work different from fact checking? If the ultimate goal is to design models that are robust to misinformation, fact-checking should be definitely considered, and models for fact-checking should be included in the evaluation. Refer to https://fever.ai/ for more details on the fact-checking literature and datasets.

3)	The setup proposed in this paper is not simulating a real-word scenario, as instead claimed in the introduction. While fake contexts can be presented, when doing retrieval to retrieve relevant documents, not all of them will contain fake context. Thus, the distribution between real/fake presented in this work does not reflect a real scenario.

Contribution of the paper.
1)	Fake detector. While in the introduction the paper claims of proposing a framework to detect against misinformation, in practice this solution is a simple combination of the score of the QA model with a fake/real classifier. This poses several limitations, including a little ability of the model to generalize when new contexts arrive. SQuAD suffers from a train test overlap problem [1]. These findings will apply to the classifier as well because it is trained on the same data. It is unclear how much the framework for contradictory QA is learning how to rely on an information, rather than memorizing because of text was already seen in train. How does the detector perform when evaluated on a set of question, paragraph and contradicting text not seen at all in train? Based on the paper, the fake contexts are perturbation of the original text, to which extend do you expect the detector working because of memoization, rather than reasoning over the text?
2)	BART-FG model. The main focus of the paper goes into this model. From the results, it remains unclear how the produced context can be contradictory, and thus what makes these contexts different from context generated for adversarial attack of QA models. This observation goes with my previous point regarding the definition of contradicting examples.

Ethical concerns.
The fact that the model is based on BART, and can be easily reproduced, is not a valid justification to release the trained model BERT-FG. Same goes with the point regarding the limited ability to generate disinformation. While every generative model can be used in theory to create fake contexts, this work describes a way of generative fake contexts, without providing (based on the results), any robust approach to mitigate the problem.

[1] Lewis at al., Question and Answer Test-Train Overlap in Open-Domain Question Answering Datasets

**Summary Of The Paper:**

This paper addresses the problem of deriving the correct answer when contradicting examples are presented to the model. First, it introduces a dataset for the task. The dataset, ContraQA is built on SQuAD, and it contains contradicting contexts produced by humans and neural-models. Then, it presents a model for generating contradicting examples. The model, BART-FG, generates fake contexts by iteratively modifying and original input paragraph. The procedure starts by applying a constituency parsing to extract constituency spans from the input sentence. Then, it randomly masks some of these constituency spans, that are eventually fill by a BART model fine-tuned on Wikipedia dump. To study how QA models behave with contradicting examples, this work evaluates the performances in a scenario where the correct and the fake contexts are presented to the model. In order to make the QA system robust to fake contexts, it proposes a misinformation-aware framework that combines the score of the model with a trust score outputs by a fake detector, which is a transformer-based model trained to classify if a context is real or fake. The results show that under this setting, model performance decreases, and that the reduction can be mitigate by applying the fake detector model. Finally, it shows a comparison, between BART-FG and GPT-2, to identify which of the two can generate more impactful fake contexts for the QA model.

**Summary Of The Review:**

The purpose idea of the dataset is valuable, as well as the problem is proposing to address. However, there are some problems that do not allow me for acceptance. This includes problem statement unclarity, claims not well supported by results, and ethical concerns.

---

### Decision · Program_Chairs · 2022-01-20

**Decision:**

Reject

**Comment:**

This paper tackles a really interesting and realistic problem: how does contradictory (potentially) fake information affect QA systems? The authors try to approach this problem by building a new dataset, starting with the widely used SQuAD and adding contradictory information. This is quite interesting, but the rest of the paper does not follow through. Reviewers ask a critical question: how would you distinguish the information that is fake, as opposed to valid, truthful information? Without this distinction, how would you train a language model to detect the fakeness and answer the question using the valid information? Unfortunately, the authors did not reply to this critical question, so it is difficult to judge the validity and contributions of this paper. There are also serious ethical implications which are discussed in the ethics review.